# Current Treatment Concepts for Extra-Abdominal Desmoid-Type Fibromatosis: A Narrative Review

**DOI:** 10.3390/cancers16020273

**Published:** 2024-01-08

**Authors:** Yong-Suk Lee, Min Wook Joo, Seung-Han Shin, Sungan Hong, Yang-Guk Chung

**Affiliations:** 1Department of Orthopaedic Surgery, Incheon St. Mary’s Hospital, College of Medicine, The Catholic University of Korea, 56 Dongsu-ro, Bupyeong-gu, Incheon 21431, Republic of Korea; maerie@catholic.ac.kr (Y.-S.L.); ydeepinyou@cmcnu.or.kr (S.H.); 2Department of Orthopaedic Surgery, St. Vincent’s Hospital, College of Medicine, The Catholic University of Korea, 93 Jungbu-Daero, Paldal-gu, Suwon-si 16247, Republic of Korea; mwjoo@catholic.ac.kr; 3Department of Orthopaedic Surgery, Seoul St. Mary’s Hospital, College of Medicine, The Catholic University of Korea, 222 Banpo-daero, Seocho-gu, Seoul 06591, Republic of Korea; tumorshin@gmail.com

**Keywords:** extra-abdominal desmoid-type fibromatosis, active surveillance, tyrosine kinase inhibitors, γ-secretase inhibitors, systemic treatment strategies, multidisciplinary management

## Abstract

**Simple Summary:**

Extra-abdominal desmoid-type fibromatosis (EADTF) is a rare but challenging condition due to its invasive growth and high recurrence rates. This comprehensive literature review, covering studies from 2008 to 2023, evaluates the currently available treatment approaches. The traditional surgical treatment for most EADTF cases has shifted to active surveillance as the initial management strategy, reflecting a more conservative and watchful approach. Surgical resection now serves as a secondary option, primarily for symptomatic patients or when tumors threaten vital structures. Systemic therapies, including tyrosine kinase inhibitors and γ-secretase inhibitors, have shown promising results in controlling disease progression. The United States Food and Drug Administration’s approval of nirogacestat as the first systemic treatment for EADTF represents a significant milestone, demonstrating marked improvements in progression-free survival and patient quality of life. The study emphasizes the necessity of individualized treatment plans and multidisciplinary management, advocating for patient-centric goals that focus on overall well-being and functional outcomes.

**Abstract:**

Extra-abdominal desmoid-type fibromatosis (EADTF) is a rare neoplastic condition of monoclonal fibroblastic proliferation characterized by local aggressiveness with a distinct tendency to recur. Although EADTF is a benign disease entity, these tumors have a tendency to infiltrate surrounding normal tissues, making it difficult to completely eliminate them without adjacent healthy tissue injury. Surgical excision of these locally aggressive tumors without clear resection margins often leads to local recurrence. The aim of this thorough review was to assess the current treatment concepts for these rare tumors. A comprehensive search of articles published in the Cochrane Library, MEDLINE (PubMed), and EMBASE databases between January 2008 and February 2023 was conducted. Surgical intervention is no longer the first-line approach for most cases; instead, strategies like active surveillance or systemic therapies are used as initial treatment options. With the exception of EADTFs situated near vital structures, a minimum of 6–12 months of active surveillance is currently advocated for, during which some disease progression may be considered acceptable. Non-surgical interventions such as radiation or cryoablation may be employed in certain patients to achieve local control. The currently preferred systemic treatment options include tyrosine kinase inhibitors, low-dose chemotherapy, and gamma-secretase inhibitors, while hormone therapy is not advised. Nonsteroidal anti-inflammatory drugs are utilized primarily for pain management.

## 1. Introduction

Extra-abdominal desmoid-type fibromatosis (EADTF) is a rare neoplastic condition of monoclonal fibroblastic proliferation characterized by local aggressiveness with a distinct tendency to recur. It is also referred to as a desmoid tumor or aggressive fibromatosis [1]. EADTF is an exceptionally infrequent disease entity, with a peak age of 30–40 years, affecting a mere 2–4 patients per 1 million of the population annually [2,3]. Although EADTF is a benign disease entity, these tumors have a tendency to infiltrate the normal surrounding tissues, making it difficult to completely eliminate them without adjacent healthy tissue injury [4]. Surgical excision of these locally aggressive tumors often leads to local recurrence [5]. EADTF originating from musculoaponeurotic structures in the upper and lower extremities presents a challenging therapeutic problem [6,7].

Although extensive resection was a desirable management strategy before the 2010s, the local recurrence rate was reported to exceed 40%, even after excision with a negative margin [8,9,10]. Thus, it is difficult to establish treatment strategies due to the high local recurrence rate, and efforts have been made in recent years to standardize the management of the disease [11].

Irrespective of the tumor site and size, the initial management strategy for newly diagnosed EADTF typically entails a judicious active surveillance approach as recent consensus [12,13]. For progressive EADTF during an active surveillance, medical intervention, radiation therapy, or surgery may be recommended, particularly for unresectable lesions near critical anatomical structures [11,13]. Medical treatment modalities encompass antihormonal therapies, nonsteroidal anti-inflammatory drugs (NSAIDs), tyrosine kinase inhibitors (TKIs), chemotherapeutic regimens, and γ-secretase inhibitors (GSIs) [14,15,16]. Thermal ablative therapies, including cryotherapy, have recently emerged as viable options for achieving local control [17].

The aim of this thorough review was to assess the current treatment concepts for these rare tumors.

## 2. Materials and Methods

### 2.1. Search Strategy

A comprehensive search of articles published in the Cochrane Library, MEDLINE (PubMed), and EMBASE databases between January 2008 and February 2023 was conducted in March 2023. We limited the search to English-written articles and removed any duplicated records. The terms used were a combination of extra-abdominal desmoid-type fibromatosis, aggressive fibromatosis, desmoid tumor, treatment, management, and therapy.

### 2.2. Inclusion and Exclusion Criteria

We included all the following reports and studies: (1) studies following patients treated for EADTF; (2) studies containing reliable endpoints, rate of disease progression, stability or regression, and progression-free survival and efficacy; (3) retrospective (*n* ≥ 30), prospective, observational, clinical trials (phase 2 or 3), systematic reviews, and consensus guidelines. The exclusion criteria were (1) studies particularly regarding intra-abdominal tumors; (2) articles including mostly biochemical or genomic outcomes; (3) analyses of <30 patients in case series studies; (4) published in the form of clinical trials (phase 1), letters, editorials, or commentaries. The inclusion and exclusion criteria are summarized in Table 1. In the case of mixed case series studies, if the number of extra-abdominal cases was less than 30, these were excluded.

The studies were selected and the data were independently extracted by two of the research authors. When they disagreed, the consensus was adjusted to reach an agreement. The data collated from each investigation included the identities of the contributing authors, the year of publication, the number of patients involved, the demographics of the patients, the research methodology, the duration of the follow-up period, and the study outcomes.

### 2.3. Study Categories

Studies were classified into seven categories based on the modality utilized. The categories were active surveillance, surgery and radiation, thermal ablative therapy, antiestrogens and NSAIDs, cytotoxic chemotherapy, TKIs, and GSIs. The analysis of outcome measures varied across different categories: the rate of disease progression, stability, or regression was analyzed in active surveillance, thermal ablative therapy, antiestrogens, NSAIDs, cytotoxic chemotherapy, TKIs, and GSIs; recurrence rate was analyzed in surgery and radiation, progression-free survival and duration of response were analyzed in all categories).

## 3. Results

### 3.1. Study Selection Process

The literature review identified 1025 potential articles. After eliminating duplicates, 562 records were obtained from the database searches. After conducting title and abstract screening, 245 publications were considered potentially relevant. Following the full-text screening, 80 articles met the inclusion criteria and were incorporated into the final review. Figure 1 illustrates a flowchart of the selection process.

### 3.2. Active Surveillance

Initial active surveillance, defined as continuous magnetic resonance imaging (MRI) 1–2 months after diagnosis and every 3–6 months, is considered the first-line approach to most patients with EADTF based on accumulated evidence pointing to long-term stabilization or spontaneous regression [2,18,19,20]. The latest two authoritative guidelines, including those from the Desmoid Tumor Working Group and the National Comprehensive Cancer Network, support active surveillance as the preferred initial treatment for most patients [11,21]. These guidelines recommend continuous imaging observation in asymptomatic patients with tumors based on anatomic locations where progression is unlikely to lead to severe morbidity, or where progression would be likely to result in significant morbidity.

Several retrospective series have supported the use of active surveillance [22,23]. A proportion of 50–70% of the patients managed with surveillance were reported to have achieved a stable condition or spontaneous regression [11,24,25,26]. Recently, three prospective and observational European studies of sporadic EADTF patients managed by active surveillance have been reported [27,28,29]. A large, prospective observational study in Italy reported that 108 patients reached a treatment-free survival rate of approximately 65% at three years, with 56% of patients undergoing spontaneous regression despite initial disease progression [27]. A Dutch study with a total of 105 patients showed 33 patients (32%) in stable condition and 29 patients (28%) with partial or complete regression [28]. A significant French trial involving 771 patients observed that 52% of the patients experienced no disease progression over two years, compared to just 25% of those who initially underwent surgery [29].

Recent guidelines propose transitioning from active surveillance to another treatment modality after a minimum of three consecutive reevaluations, and potentially after at least one year since the initial diagnosis [11].

### 3.3. Surgery and Radiotherapy

Until the early 2000s, surgical resection with negative margins had been the mainstay of EADTF treatment, similar to other soft tissue sarcomas. However, local recurrence rates after surgical treatment at 5–10 years have been reported up to 60% [10,30,31,32]. A prospective largest cohort study of 771 patients comparing initial surgery to initial surveillance demonstrated no difference between the surgery group and surveillance group in event-free survival which was slightly over 50% (53% vs. 58%) [29]. Moreover, in an analysis of 216 patients, no differences in overall survival were found at 5 and 10 years compared with patients who underwent front-line surgery and were managed by active surveillance [33].

The European Pediatric Soft Tissue Sarcoma Study Group (EpSSG) enrolled 173 pediatric patients. A total of 54 underwent active surveillance and 47 received immediate surgery. The study revealed that the five-year progression-free survival rate was 26.7% in the observation group and 41.2% in the surgery group [34].

The association between negative margins and a decrease in the rate of recurrence has been subject to debate. Resections with microscopically negative margins were not achieved in most surgeries, and there was no consensus on whether a positive margin resection correlated with the increased risk of recurrence [35]. Similarly, an analysis of a series of 495 patients with EADTF who underwent surgery did not find any significant association between the status of surgical margins (R0 vs. R1) and the risk of recurrence [10].

Even though surgical resection is no longer a primary option based on its high local recurrence rate, surgery remains a considerable option for symptomatic patients or those with progressive EADTF [36,37]. When the disease advances, the initial treatment options include surgical intervention or systemic therapies, depending on the location of the tumor. Surgical procedures may be prioritized for tumors endangering vital anatomy, while for the majority of situations, surgery is reserved as a subsequent option following the ineffectiveness of systemic approaches, which encompass chemotherapy and targeted molecular treatments [11,38].

Radiation therapy is presently considered as treatment following surgery or systemic therapies, especially in cases where surgical procedures pose a significant risk of complications [20]. The reported local control rates are advantageous following radiation alone and range from 55 to 92.3% [39,40,41,42].

In a recent cross-sectional cohort analysis involving 412 patients diagnosed with EADTF, 109 patients received radiation alone and 85 were treated with a combination of surgery and radiotherapy [43]. The median dose for patients treated with radiation alone was 56 Gy (range, 56–75 Gy), whereas it was 50.4 Gy (range, 50–66 Gy) for combined therapy. The radiation alone group had a five-year local control rate of 65%, compared to 77% in the surgery and radiation-combined group. A phase II trial enrolled 44 patients and focused on the effects of definitive radiation therapy at a moderate dose alone for patients with EADTF, which was either inoperable or progressing [44]. The study encompassed individuals with initial, recurrent, or partially removed EADTFs who underwent a course of 56 Gy in 28 fractions. With a three-year local control rate of 81.5% and a median follow-up of 4.8 years, complete remission was achieved in 13.6% of cases, while a partial response was seen in 36.4%, and stable disease was noted in 40.9%.

The use of adjuvant radiotherapy is not recommended for pediatric patients with EADTF due to its associated risks, particularly the potential for radiation-induced sarcoma in this young patient population [11].

### 3.4. Thermal Ablative Therapy

Recently, image-guided percutaneous ablation, including cryotherapy, has been a popular option for the treatment of EADTF [17,45]. The procedure uses argon gas through a sealed, segmentally insulated probe to cause rapid cooling. The formation of ice within and outside of cells by cryoablation leads to tissue necrosis through the direct destruction of cell membranes, osmotic dehydration, and membrane rupture, in addition to causing vascular damage and thrombosis. This modality has been revealed to be effective for both first-line treatment and disease recurrence after prior therapy. A total of 22 cryoablation patients were matched with 33 surgical patients and two-year disease control was 85% [46]. Thirty-four patients with 41 lesions were treated with cryoablation, with 42.2% lacking pain and no progression on imaging at three years [47]. The only prospective phase II trial including 50 patients was reported to be a highly effective means of local tumor control with improved pain control after ablation. The non-progression rate at 12 months was 85.6%, including a complete response in 28.6%, partial response in 26.2%, and stable disease in 31% [17]. All studies of other ablative therapies, such as high intensity focused ultrasound and microwave ablation, were excluded due to sample sizes of less than 30.

### 3.5. Antiestrogens and Nonsteroidal Anti-Inflammatory Drugs

Estrogen has long been hypothesized to modulate EADTFs; hence, antihormonal agents such as tamoxifen or toremifene have been frequently used to treat EADTFs [48]. Several retrospective case series including 146 patients, 44 patients, and 32 patients reported a wide range of disease control rates (25–89.6%) [11,49,50,51]. The only prospective phase II study evaluating antihormonal therapy plus NSAIDs showed 36% of two-year progression-free survival and survival rates [52]. A recent study showed no definite relationship between size, MRI signal, and symptom improvement during tamoxifen treatment [51]. Consequently, there is a lack of proof regarding antihormonal treatments in patients with EADTF, and current clinical guidelines have ceased to support hormone therapies as a standard recommendation [11,21].

The use of NSAIDs to manage EADTFs was initiated based on the understanding that there is an overexpression of COX-2 in these tumors [53]. Various response rates have been reported [54,55]. A prospective study of 31 patients with meloxicam treatment reported 35.5% with progressive disease [55]. NSAIDs are presently not considered agents for disease management, and current guidelines recommend their use solely for pain control [11].

### 3.6. Cytotoxic Chemotherapy

Chemotherapy combining low doses of methotrexate (MTX) and vinblastine (VBL), referred to as low-dose chemotherapy, is generally administered and associated with disease control rates of 64–100% [56,57]. The standard doses for these treatments are MTX at 30 mg/m^2^ and VBL at 6 mg/m^2^, with treatment cycles varying from once a week to every four weeks. A prospective phase II clinical trial involving 37 patients demonstrated a 95% clinical benefit rate and an 80.8% rate of progression-free survival at five years [58]. Another retrospective report treating 75 patients showed a 99% disease control rate at a median administered duration of 14 months [59]. Chemotherapy using low doses of MTX and vinorelbine (VNL) also demonstrates high efficacy, with a clinical benefit rate of 98% [60].

It is recommended that treatment should last for at least one year, with more than 40 cycles being ideal, and therapy should be resumed in the event of a recurrence. Typically, the therapeutic effects may not be perceptible for several months; yet, the benefits are likely to persist substantially past the cessation of active treatment [56,59].

Similar results were observed in pediatric patients. A European study reported a difference in the five-year progression-free survival rate between the observation group (*n* = 54) and the chemotherapy group (*n* = 53), which were 27% and 43%, respectively, although this difference was not statistically significant [34].

### 3.7. Tyrosine Kinase Inhibitors

Tyrosine kinase inhibitors are extensively employed in managing EADTF, especially in cases of progressive, refractory, or critical tumors, due to their potentially rapid therapeutic action [61].

Numerous prospective trials have evaluated the effectiveness of TKIs (imatinib, pazopanib, and sunitinib) in patients with EADTF [62,63,64,65,66]. In a double-blind phase III, placebo-controlled, randomized trial involving 87 patients with progressive EADTFs, results were reported for those who randomly received either sorafenib (400 mg per day) or placebo [66]. The sorafenib cohort exhibited a 71% two-year progression-free survival rate, whereas the placebo cohort had a rate of 36%. The progression of tumor occurred in only six patients (12%). Another prospective observational study on sorafenib enrolled 104 patients with a 46.1% objective response rate and 31.7% with stable disease [67]. The progression-free survival rate at one and two years was 86.6% and 73.7%, respectively. In a phase II trial conducted by the French Sarcoma Group, which was prospective, open-label, and randomized, 72 patients with progressive EADTF were randomly allocated for treatment with either pazopanib or MTX-VBL. The outcome indicated an improved rate of objective response (37% compared to 25%) and one-year progression-free survival (86% vs. 67%) [64]. The tumor progression-free survival at six months was 83.7%, similar to that of sorafenib.

Additional research is required to determine the most effective dosage, treatment duration, and order of administration for TKIs to ascertain their role in managing EADTF more accurately. Recent guideline declared that current evidence demonstrated a significant therapeutic advantage of TKIs in the treatment of EADTF and recommended them as a systematic management option for patients with progressive EADTF [11].

The study on TKIs involved 51 patients aged ten years and older, which included a pediatric population. The progression-free survival rates at two months, four months, one year, and three years were 94%, 88%, 66%, and 58%, respectively [61].

### 3.8. γ-Secretase Inhibitors

A selective GSI, nirogacestat, has been shown to impede cell growth causing cell cycle arrest, suggesting the possible application of GSIs in EADTF [68].

In a phase I trial of nirogacestat, five out of seven patients (71.4%) with EADTF experienced a partial response, while the remaining two achieved stable disease, culminating in a disease control rate of 100% [69,70]. Subsequently, an open-level phase II study involving 17 patients with EADTF who were treated with other prior therapy was conducted. In this study, there were 5 partial responses (29%) and 12 patients (71%) with stable disease, which also resulted in a disease control rate of 100% [71]. Symptoms were considerably reduced in patients who achieved a partial response. A phase III, randomized, double-blinded, and placebo-controlled trial, involving nirogacestat, enrolled 142 patients, dividing them into two groups: 70 received nirogacestat and 72 were given placebo. The study revealed a significant improvement in progression-free survival among those with EADTF. At the two-year mark, 76% of the patients treated with nirogacestat remained event-free compared to 44% in the placebo group. Complete response rates were 7% for the nirogacestat cohort and 0% for the placebo cohort [16].

The RINGSIDE phase III trial, a double-blind, placebo-controlled study, is assessing the efficacy of the γ-secretase inhibitor AL102 at a dosage of 1.2 mg once daily. This regimen, selected from phase II results, is for adults and adolescents aged 12 or older with recurrent or newly diagnosed progressing desmoid-type fibromatosis, as determined by the investigator. The aim is to enroll over 156 subjects internationally, randomizing them at a 1:1 ratio to receive either AL102 or a placebo daily. The primary endpoint of the trial is progression-free survival [72].

Tegavivint, an inhibitor of the Wnt and beta–catenin pathway, is currently being tested as a new therapeutic agent in a phase II trial with five pediatric patients [73].

Table 2 summarizes current concepts recommended for the treatment of ex-tra-abdominal desmoid-type fibromatosis.

## 4. Discussion

The approach to treating EADTF has undergone significant changes over the past ten years. Surgical intervention is no longer the first-line approach for most cases; instead, strategies like active surveillance or systemic therapies are used as initial treatment options [31,74,79]. Guidelines from the Desmoid Tumor Working Group and the National Comprehensive Cancer Network suggest that most patients should begin with active surveillance and undergo ongoing monitoring [11,21]. Continuous monitoring with serial MRI is considered for active surveillance [11]. Patients should be scheduled for initial surveillance within 1–2 months of diagnosis and then at 3–6-month intervals. Active treatment should be contemplated in cases where the tumor continues to progress, symptoms intensify, or the risk of disease morbidity escalates. These decisions should be based on at least two additional evaluations and generally do not apply within the first year of diagnosis. This is due to the potential for EADTF to undergo spontaneous regression after initial advancement [80,81]. Nevertheless, if the tumor is located adjacent to a vital anatomical structure that could significantly increase morbidity, it may be appropriate to consider proactive treatment earlier to mitigate the potential risk of morbidity before the disease stabilizes [74].

Surgery is typically not the first-line treatment option for EADTF affecting the extremities, except under particular conditions approved by a multidisciplinary tumor board. When considering surgical intervention, preserving function is the primary objective. Other than surgery, both radiotherapy and medical therapy can be effective alternative treatment strategies for this site [11,12].

While achieving microscopic negative margins is the goal, there are instances where positive microscopic margins might be deemed tolerable to preserve function or cosmetic appearance. When a positive resection margin occurs during initial management, current evidence is inadequate to support the routine use of adjuvant radiotherapy or additional surgical intervention [36]. Radiotherapy may be an option for EADTF in scenarios where local recurrence would pose considerable surgical challenges and potentially result in substantial morbidity [82].

Systemic therapy, including low-dose chemotherapy, conventional chemotherapy, or TKIs (sorafenib, imatinib, pazopanib, and sunitinib) is utilized for cases presenting with rapidly progressing and symptomatic inoperable tumors or advanced disease. [14]. In this study, chemotherapy regimens with other agents, such as doxorubicin-based cycles or hydroxyurea were excluded due to their sample sizes being less than 30.

Notch signaling and dysregulation between the Notch and Wnt/β-catenin signaling pathways, in relation to oral GSIs, play a role in the tumorigenesis, progression, and treatment resistance of the disease [68,83]. This medical treatment has demonstrated improvements in progression-free survival rates in phase III trials as a novel treatment option [16]. On 27 November 2023, the United States Food and Drug Administration approved nirogacestat (Ogsiveo) for adults with advancing desmoid-type fibromatosis that requires systemic treatment, marking it the first approved treatment for these tumors. This approval was based on the results from the phase 3 DeFi clinical trial mentioned above. Nirogacestat showed significant improvements in clinical efficacy in the phase 3 DeFi trial, substantially improving pain, symptom burden, physical and role functioning, and health-related quality of life in desmoid tumor patients, all while maintaining a manageable safety profile. Consequently, it is highly probable that the trend of initial systemic treatment for extra-abdominal desmoid-type fibromatosis will shift towards nirogacestat in the future.

All studies of FAP associated EADTF were subject to exclusion criteria and, therefore, were not included in this study. However, we added the content about FAP in the discussion section as it would be relevant to pediatric patient population. No single study was exclusively focused on treatment of EADTF; however, there were mixed studies that included eight cases in 2017, three cases in another study from 2017, and three cases in 2011 [84,85,86]. FAP-associated desmoid tumors are predominantly located intra-abdominally and in the abdominal wall, while sporadic desmoid tumors are usually found in the extra-abdominal area [87]. FAP-associated desmoid tumor seems to be characterized by a more aggressive clinical course. Nevertheless, the overall treatment concept for FAP-associated desmoid tumors does not differ from that for non-FAP-associated desmoid tumors, although they are often treated with more aggressive systemic treatments [88]. The same considerations can also be applied to patients with FAP-associated desmoid tumors, with similar 10-year progression-free survival (33% for the wait-and-see approach and 49% for surgery) and better outcomes for extra-abdominal and abdominal wall desmoid tumors [86].

There are several limitations to this review. First, as EADTF is a rare disease, the level of evidence in most studies was low. Second, because there is a lack of reports on treatment-associated functional impairment, we were only able to evaluate the clinical outcomes of patients with postoperative complications. Third, most of the literature, except the most recent studies, did not describe details of symptoms. Thus, an analysis of symptomatic improvement could not be performed.

Extra-abdominal desmoid-type fibromatosis, which is a rare condition, has a highly variable presentation and symptoms, necessitating highly individualized treatment approaches. Mandatory case-by-case discussions by experienced multidisciplinary tumor board members are essential. The main goals of treatment should extend beyond clinical markers, such as progression-free survival, to consider the overall wellbeing of the individual, including pain, functional ability with daily activities, and the overall quality of life. Surveillance should be considered the initial step at the time of diagnosis whenever possible. Any treatment approach—whether medical, regional, or surgical—must be carefully selected through balancing the benefits and the risks.

## 5. Conclusions

Active surveillance should be considered the first-line management approach for EADTF, with treatment individualized based on the case’s complexity and progression. Surgical options, while less favored as an initial treatment, remain relevant for symptomatic patients or when the tumor endangers critical anatomy. Systemic therapies, particularly TKIs and GSIs, are emerging as effective treatments for progressive and symptomatic EADTF. This review underscores the importance of a multidisciplinary approach to managing EADTF, prioritizing patient well-being and the quality of life alongside clinical outcomes.

## Figures and Tables

**Figure 1 cancers-16-00273-f001:**
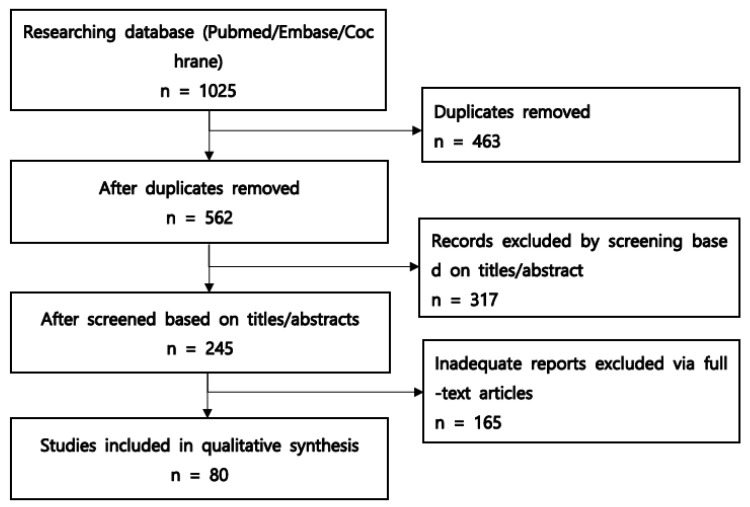
Flow diagram of the literature search and selection process.

**Table 1 cancers-16-00273-t001:** Inclusion and exclusion criteria.

	Inclusion Criteria	Exclusion Criteria
Population	Patients with extra-abdominal desmoid-type fibromatosis	Patients with intra-abdominal desmoid-type fibromatosis Patients with other neoplasms
Intervention and Comparators	Active surveillance Surgery Radiation Thermal ablative therapy Antiestrogens and anti-inflammatory drugs Cytotoxic chemotherapy Tyrosine kinase γ-secretase inhibitors	
Outcomes	Type of treatment modalities Rate of disease progression, stable, or regression Progression-free survival period Recurrence rate	Biochemical, molecular, or genetic outcomes
Study design	Case series studies (*n* ≥ 30) Prospective, observational studies Clinical trials (phase 2 or 3) Systematic reviews/meta-analyses Consensus or practice guidelines	Small sample size (*n* < 30) Clinical trials (phase 1) Guidelines or reviews before 2008 Letter, editorial, commentary

**Table 2 cancers-16-00273-t002:** Current concepts recommended for the treatment of extra-abdominal desmoid-type fibromatosis.

Treatment Option	Application	Outcome and Efficacy	Side Effects
Active surveillance	All guidelines recommend schedule for initial surveillance within 1–2 months of diagnosis and then at 3–6-month intervals [11,21,74].	SD, 59%; PR, 19%; PD, 20% [18]. DC, 60–82% [75]. 2-year event-free survival, 58% [29]. SD, 36%; SR, 27% [23]. SR, 28.4% (follow-up, 32 months) [24]. SD, 65%; PR, 25%; SR, 5% (follow-up, 35.7 months) [25]. SR, 25%, PD, 39% (follow-up, 32.3 months) [27]. SD, 32%, SR, 28%, PD, 40% [28].	
Surgery	Surgery is typically not the first-line treatment option, except under particular conditions approved by a multidisciplinary tumor board [21]. When considering surgical intervention, preserving function is the primary objective [2].	Recurrence rates: positive margin, 32%; negative margin, 40% [30]. Recurrence rates: 14–47.2% [32,35,36,76,77]. Risk of local recurrence with microscopically positive margins: risk ratio, 1.78 [36].	Surgical complications, the need for complex surgical reconstruction, and decreased quality of life [2].
Radiotherapy	Postoperative RT or administered alone; 56–60 Gy in 28 fractions [44,78].	DC: 55–92.3% [39,40,41,42]. 3-year DC: 81.5% (SD, 40.9%; PR, 36.4%; CR, 13.6%) [44]. 5-year DC with RT ± surgery: 77% and 65% [43].	Fibrosis, fracture, and secondary malignancy [43,44].
Cryoablation	Two 10 min freeze–thaw cycles [17,46].	SD: 31%; PR: 26.2%; CR: 28.6% at 12 months [17]. 2-year DC: 85% [46]. 3-year DC: 42.2% [47].	Nerve injury, rhadomyolysis, skin necrosis, bleeding, infection, and colo-cutaneous fistula: 2.4–30% [17].
Antiestrogens	Lack of proof to regard antihormonal treatments, and current clinical guidelines have ceased to support hormone therapies as a standard recommendation [11,21].	Wide range of DC (25–89.6%) [11,49,50,51]. Antihormonal therapy + NSAIDs: 36% of 2-year PFS [52]. No correlation between size and MRI signal changes and symptom release [51].	
Anti-inflammatory drugs	Not considered as agents for disease management, and current guidelines recommend their use solely for the control of pain [11].	Various response rates [54,55]. Prospective studies with meloxicam: PD, 35.5% [55].	No life-threatening toxicity [52].
Cytotoxic chemotherapy	MTX (30 mg/m^2^) and vinblastine (5 or 6 mg/m^2^) every 7–10 days. Weekly MTX (30 mg/m^2^) and vinorelbine (20 mg/m^2^); 40 to 50 cycles.	DC: 64–100% [56,57,59]. 1-year PFS: 79% [64]. SD, 17%; PR, 39%; CR, 42% [59].	Hematologic toxicity: bone marrow suppression. Nonhematologic toxicity: nausea, vomiting [59].
Tyrosine kinase inhibitors	200 to 800 mg of oral Imatinib daily. 37.5 to 52 mg of Sunitinib daily dose. 400 mg of Sorafenib daily dose. 800 mg of Pazopanib daily dose.	CR + PR: 2–6% (between 3–6 months) [61,62,63]. 1-year PFS: 66% [63]. DC: 68.4% [65]. 2-year PFS: 74.7% [65]. 1-year PFS: 86.6–89% [66,67]. 6-month PFS: 83.7% [64].	Hematologic toxicity: neutropenia [65]. Nonhematologic toxicity: fatigue, diarrhea, nausea, weight loss, hypertension, hand-foot skin reaction, rash, alopecia [64,65,66,67].
γ-secretase inhibitors	150 mg twice daily of Nirogacestat (NCT03785964; DeFi trial). 1.2–4 mg daily of AL 102 (RINGSIDE phase 2/3 trial).	2-year event-free: 76% CR, 7% [16]. Endpoints: PFS, Overall Response Rate, Duration of Response, Quality-of-life measures [72].	Diarrhea, nausea, fatigue, hypophosphatemia, maculopapular rash [16].

CR, complete response; DC, disease control; MTX, methotrexate; NSAID, nonsteroidal anti-inflammatory drug; PD, progressive disease; PFS, progression-free survival; PR, partial response; RT, radiotherapy; SD, stable disease; SR, spontaneous regression.

## Data Availability

The data presented in this study are available on request from the corresponding author.

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
