# Peer review of "Current Treatment Concepts for Extra-Abdominal Desmoid-Type Fibromatosis: A Narrative Review"

_cancers, 2024, doi:10.3390/cancers16020273_

Round 1

Reviewer 1 Report

Comments and Suggestions for Authors

Dear Authors,

thank you very much for this comprehensive and timely review about management strategies for extraabdominal desmoid tumors. Even though the paper does not add new findings, it is a very good and concise overview of the current treatment startegies for desmoid tumor patients.

As this manuscript is very much in line with the current recommendations from the globally acting Desmoid Tumor Working Group I have absolutely no objections.

Thanks again for this concise overview and kind regards.

Author Response

Dear Authors,

thank you very much for this comprehensive and timely review about management strategies for extraabdominal desmoid tumors. Even though the paper does not add new findings, it is a very good and concise overview of the current treatment startegies for desmoid tumor patients.

As this manuscript is very much in line with the current recommendations from the globally acting Desmoid Tumor Working Group I have absolutely no objections.

Thanks again for this concise overview and kind regards.

-> Thank you for your comprehensive review and comment.

Reviewer 2 Report

Comments and Suggestions for Authors

This is a well written comprehensive review of management of extra abdominal desmoid tumors. Following points would be helpful to address:

1) Line 132 does not read well. Consider revising it

2) Doxil chemotherapy has been shown to be effective in a study comprising of > 30 patients (reference Davis et al ASCO 2022 abstract 11581)  . Would highly recommend including it as part of review. 

3) Now with FDA approval of nirogacestat how do the authors propose a a decision be made regarding the choice of initial systemic treatment - TKI vs GSI?

4) Would be also helpful to the readers to include discussion about ongoing trials of newer agents targeting extra abdominal desmoid tumor patients > 30 in number.  

5) Should FAP associate extra abdominal desmoid tumors be managed differently as compared to non-FAP associated desmoids given their relatively aggressive disease biology? 

6) How does age of the patient impact the use of treatment strategies? Did any of the trials/studies reviewed include pediatric patients? If so then those should be specifically highlighted as it is being submitted to peds oncology section. 

Author Response

Thank you for your thoughful review. We revised the paper to reflect your comments and the details are as follows.

1) Line 132 does not read well. Consider revising it

--> Thank you for the advice. We revised Line 132 to read well.

2) Doxil chemotherapy has been shown to be effective in Davis ASCO 2022 doxil chemotherapya study comprising of > 30 patients (reference Davis et al ASCO 2022 abstract 11581)  . Would highly recommend including it as part of review.

--> Thank you for the suggestion. The study that mentioned above is a retrospective study includes 40 cases, however only 19 cases are extra-abdominal and thus meet the exclusion criteria. There have been several studies on chemotherapy based on pegylated liposomal doxorubicin. While these studies report that it is well-tolerated, with less cardiac toxicity than conventional doxorubicin and an effective systemic therapy, they did not meet the inclusion criteria for this analysis. But would you put it in the discussion as an exception? Please understand that, in order to increase the credibility of the retrospective results, the number of enrolled patients was set to 30.

3) Now with FDA approval of nirogacestat how do the authors propose a a decision be made regarding the choice of initial systemic treatment - TKI vs GSI?

--> You must have suggested a good point. On November 27, 2023, the U.S. Food and Drug Administration (FDA) approved nirogacestat (Ogsiveo) for adults with advancing desmoid-type fibromatosis that requires systemic treatment, marking it as the first approved treatment for these tumors. This approval was based on the results from the international, multicenter, randomized, double-blind, placebo-controlled, phase 3 DeFi clinical trial. The trial enrolled 142 patients with inoperable, progressing desmoid-type fibromatosis. The study revealed a significant improvement in progression-free survival. At the two-year mark, 76% of the patients treated with nirogacestat remained event-free compared to 44% in the placebo group. Complete response rates were 7% for the nirogacestat cohort and 0% for the placebo cohort.

The nirogacestat showed significant improvements in clinical efficacy in the phase 3 DeFi trial, substantially improving pain, symptom burden, physical and role functioning, and health-related quality of life for desmoid tumor patients, all while maintaining a manageable safety profile. Consequently, it is highly probable that the trend of initial systemic treatment for extra-abdominal desmoid-type fibromatosis will shift towards nirogacestat in the future. We added it in the Discussion sections.

4) Would be also helpful to the readers to include discussion about ongoing trials of newer agents targeting extra abdominal desmoid tumor patients > 30 in number. 

--> As you recommended, we have incorporated ongoing trials involving novel agents. The RINGSIDE Phase III trial, a double-blind, placebo-controlled study, is assessing the efficacy of the γ-Secretase Inhibitor AL102 at a dosage of 1.2 mg once daily. This regimen, selected from Phase II results, is for adults and adolescents aged 12 or older with recurrent or newly diagnosed progressing desmoid-type fibromatosis, as determined by the investigator. The aim is to enroll over 156 subjects internationally, randomizing them in a 1:1 ratio to receive either AL102 or a placebo daily. The primary endpoint of the trial is progression-free survival.  Tegavivint, an inhibitor of the Wnt and beta-catenin pathway, is currently in a Phase II trial with 5 pediatric patients being tested as a new therapeutic agent. We added it in the Result sections.

5) Should FAP associate extra abdominal desmoid tumors be managed differently as compared to non-FAP associated desmoids given their relatively aggressive disease biology?

--> Thank you for the suggestion. All studies of FAP associated extra-dominal desmoid tumor were subject to exclusion criteria and, therefore, were not included in this study. No single study was exclusively focused on extra-abdominal desmoid-type fibromatosis; however, there were mixed studies that included 8 cases in 2017, 3 cases in another study from 2017, 3 cases in 2011, and 14 cases in 2008. FAP-associated desmoid tumors are predominantly located intra-abdominally and in the abdominal wall, while sporadic desmoid tumors are usually found in the extra-abdominal area. As you mentioned above, FAP-associated desmoid tumor seem to be characterized by a more aggressive clinical course. Nevertheless, the overall treatment concept for FAP-associated desmoid tumors does not differ from that for non-FAP-associated desmoid tumors, although they are often treated with more aggressive systemic treatments. The same considerations can also be applied to patients with FAP-associated desmoid tumors, with similar 10-year progression-free survival (33% for the wait-and-see approach and 49% for surgery) and better outcomes for extra-abdominal and abdominal wall desmoid tumors. Do you want me to include it in Discussion section?

6) How does age of the patient impact the use of treatment strategies? Did any of the trials/studies reviewed include pediatric patients? If so then those should be specifically highlighted as it is being submitted to peds oncology section.

--> Thank you for your suggestions. Pediatric patients were reviewed in various studies, showing that their overall concept of treatment is similar to that of adults. Their therapeutic strategies should follow the proposed treatment algorithm according to current data. However, radiotherapy is not recommended for radiation-induced sarcoma. These contents were added to each part of the Result section.

If you give us another piece of advice, we will happily consider the next revision. We look forward to your affirmative reply.

Thank you for your time and consideration.

Reviewer 3 Report

Comments and Suggestions for Authors

Very nice overview and well-timed regarding the latest studies on new systemic therapies for desmoid type fibromatosis. This review is exhausting and well performed, i did not miss any of the important desmoid papers. 

One thing I would be carefull with is stating that a minimum of 6 to 12 months of active surveillance is advocated, while major treatment decisions should not be taken within the first year of diagnosis (as you write yourself at page 8 phrase 280). 

I would like to see emphasised in the discussion that desmoid type fibromatosis is a rare conditions and it is advised to leave decisions on treatment to an experienced multidisciplinary tumor board.

Comments on the Quality of English Language

There are just some minor comments, for example:

- page 9 phrase 307, it should be "Nirogacestat" not "The Nerogasestat".

I think it might be a good idea to ask for an English proof-reading of the manuscript to tackle the small errors in English language.

Author Response

Thank you for your affirmative review on our manuscript again. We have described additional content to reflect your point and the details are as follows.

Very nice overview and well-timed regarding the latest studies on new systemic therapies for desmoid type fibromatosis. This review is exhausting and well performed, i did not miss any of the important desmoid papers.

One thing I would be carefull with is stating that a minimum of 6 to 12 months of active surveillance is advocated, while major treatment decisions should not be taken within the first year of diagnosis (as you write yourself at page 8 phrase 280).

I would like to see emphasised in the discussion that desmoid type fibromatosis is a rare conditions and it is advised to leave decisions on treatment to an experienced multidisciplinary tumor board.

--> As you commented, we added the content mentioned above in the Discussion and Conclusion section.

Comments on the Quality of English Language

There are just some minor comments, for example:

- page 9 phrase 307, it should be "Nirogacestat" not "The Nerogasestat".

--> I tried to find “The Nerogasestat” in page 9 phrase 307, but I cannot find that word in the manuscript. If there is anything I am missing, I am willing to correct it upon notification.

I think it might be a good idea to ask for an English proof-reading of the manuscript to tackle the small errors in English language.

--> Thank you for your suggestion. We have received English proofreading to correct the language.

If you additionally recommend any better changes, we will gladly revise our manuscript. We look forward to your favorable reply.

Thank you for your time and consideration.

Reviewer 4 Report

Comments and Suggestions for Authors

This review paper assesses the current management used in the treatment of extra-abdominal desmoid-type fibromatosis (EADTF). The authors clarified the methods used in treating EADTF into seven categories and described each based on the information collected from searching the literature.

There are substantial parts in the paper that repeat each other, e.g., the content in the section Simple Summary is the same as in the Conclusion and the second half of the Abstract. Authors should reconsider these parts and their contribution to the article.

It is good that the authors tried to summarize the methods or treatment options used in the current management of EADTF in Table 2. However, instead of using “Efficacy”, “Outcome” will be more accurate in terms of whatever is described in the Table. It would be even better if the authors could list the efficacy and the side effects of each method used in Table 2.

Keywords should be the words that are significantly highlighted in the paper, not any general terms. Words like treatment, therapy and management are generally not used as keywords.

The English writing needs to be polished.

Comments on the Quality of English Language

The English writing needs to be polished.

Author Response

Thank you for your thoughful review. We revised the paper to reflect your comments and the details are as follows.

There are substantial parts in the paper that repeat each other, e.g., the content in the section Simple Summary is the same as in the Conclusion and the second half of the Abstract. Authors should reconsider these parts and their contribution to the article.

--> Thank you for your considerable advice. We revised Simple Summary as you suggested.

It is good that the authors tried to summarize the methods or treatment options used in the current management of EADTF in Table 2. However, instead of using “Efficacy”, “Outcome” will be more accurate in terms of whatever is described in the Table.

--> Thank you for your suggestion. We change efficacy to outcome in the Table 2.

It would be even better if the authors could list the efficacy and the side effects of each method used in Table 2.

--> Thank you for the advice. We added an efficacy section to the outcome part and also created a new section for side effects in Table 2.

Keywords should be the words that are significantly highlighted in the paper, not any general terms. Words like treatment, therapy and management are generally not used as keywords.

--> Thank you for your advice. We changed keywords in the manuscript as you recommend.

The English writing needs to be polished.

--> Thank you for your suggestion. We have received English proofreading to correct the language.  

If you give us another piece of advice, we will happily consider the next revision. We look forward to your affirmative reply.

Thank you for your time and consideration.

Reviewer 5 Report

Comments and Suggestions for Authors

Thank you for your manuscript.

I have one major question and some minor suggested edits.

1. The review process includes going from 1025 publications to 80 of interest. It is not clear to me how this process worked. You went from 245 articles which were considered potentially relevant to 80 that met inclusion criteria and were incorporated. However, a quick pubmed search did note several articles which met inclusion criteria listed in your methods but were not included in the review. Were other criteria used to eliminate articles?

Minor edits

Line 49-50 - I think you inadvertently repeated the same line or portion thereof

52 - you note surgical resection without clear margins often leads to local recurrence; however, later in the manuscript you note this is controversial as many still recur with negative margins - perhaps I would suggest 'surgical excision often leads to local recurrence' in this sentence as you then later discuss the issue of margins in more detail

63 - I don't think 'surgical procedures' fits in this sentence when you mention 'particularly for unresectable lesions'. Consider revising sentence.

108 -I would suggest replacing 'researches' - ? publications or other

129 -'56% of patients underwent spontaneous remission' - do you mean 'regression'?

186 - although I agree this publication describe a 'disease-free' group, in their results no patient is actually disease-free. Rather they describe DFS to be the time between cryoablation the recurrence on imaging data according to Recist 1.1. This is not the traditional definition of DFS or accurate as in their results all patients have residual disease on imaging (none are truly disease free).  Rather they are describing disease free as lack of pain and no progression on imaging. I would suggest clarifying in your manuscript.

279 - replace 'shoulder' with 'should be'?

Comments on the Quality of English Language

Minor issues as noted above.

Author Response

Thank you for your thoughful review. We revised the paper to reflect your comments and the details are as follows.

I have one major question and some minor suggested edits.

  1. The review process includes going from 1025 publications to 80 of interest. It is not clear to me how this process worked. You went from 245 articles which were considered potentially relevant to 80 that met inclusion criteria and were incorporated. However, a quick pubmed search did note several articles which met inclusion criteria listed in your methods but were not included in the review. Were other criteria used to eliminate articles?

--> Thank you for the suggestion. One of the most difficult criteria for inclusion is that it should include at least 30 cases of extra-abdominal cases. Most of the retrospective studies involving more than 30 cases include all intra-abdominal, abdominal wall and extra-abdominal tumors, but there are often fewer than 30 cases in terms of extra-abdominal tumors alone. We have added that content more clearly to the manuscript.

Minor edits

Line 49-50 - I think you inadvertently repeated the same line or portion thereof

--> Thank you for the advice. As you suggested, we deleted the repeated line.

52 - you note surgical resection without clear margins often leads to local recurrence; however, later in the manuscript you note this is controversial as many still recur with negative margins - perhaps I would suggest 'surgical excision often leads to local recurrence' in this sentence as you then later discuss the issue of margins in more detail

--> As you suggested, we followed your comment.

63 - I don't think 'surgical procedures' fits in this sentence when you mention 'particularly for unresectable lesions'. Consider revising sentence.

--> Following your comment, we changed to ‘surgery’

108 -I would suggest replacing 'researches' - ? publications or other

--> As you suggested, we replaced that word.

129 -'56% of patients underwent spontaneous remission' - do you mean 'regression'?

--> Thank you for your advice. We corrected.

186 - although I agree this publication describe a 'disease-free' group, in their results no patient is actually disease-free. Rather they describe DFS to be the time between cryoablation the recurrence on imaging data according to Recist 1.1. This is not the traditional definition of DFS or accurate as in their results all patients have residual disease on imaging (none are truly disease free).  Rather they are describing disease free as lack of pain and no progression on imaging. I would suggest clarifying in your manuscript.

--> Thank you for your advice. We described as you suggested.

279 - replace 'shoulder' with 'should be'?

--> As you suggested, we corrected.

If you give us another piece of advice, we will happily consider the next revision. We look forward to your affirmative reply.

Thank you for your time and consideration.

Round 2

Reviewer 2 Report

Comments and Suggestions for Authors

Thank you making the suggested changes and your response to my comments. Yes the comment about FAP should be added in the discussion section as it would be relevant to pediatric patient population. As regards to Doxil the data can be left out since 30 is the required criteria as per the study. 

Author Response

Thank you for the detailed feedback in the review. We have updated the manuscript in response to your comments and the details are as follows.

Thank you making the suggested changes and your response to my comments. Yes the comment about FAP should be added in the discussion section as it would be relevant to pediatric patient population. As regards to Doxil the data can be left out since 30 is the required criteria as per the study.

-> Thank you for your suggestion. We added the contents about FAP associated desmoid-type fibromatosis in the Discussion section.

If you give us any additional suggestions, we are eager to consider the next revision. We anticipate your positive response with great interest.

Thank you for your time and consideration.

Reviewer 4 Report

Comments and Suggestions for Authors

Moderate editing of the English language will improve the article significantly. 

Comments on the Quality of English Language

Accept after the editing of language.

Author Response

Thank you for your feedback in the review.

Moderate editing of the English language will improve the article significantly.

à Thank you for the advice. This manuscript has been edited by professional English editing services twice. The certificate is attached.

If you give us any additional suggestions, we are eager to consider the next revision. We anticipate your positive response with great interest.

Thank you for your time and consideration.
